# Accelerated Corrosion Tests in Quality Labels for Powder Coatings on Galvanized Steel—Comparison of Requirements and Experimental Evaluation

**DOI:** 10.3390/ma14216547

**Published:** 2021-11-01

**Authors:** Izabela Kunce, Agnieszka Królikowska, Leszek Komorowski

**Affiliations:** Road and Bridge Research Institute, 03302 Warsaw, Poland; akrolikowska@ibdim.edu.pl (A.K.); lkomorowski@ibdim.edu.pl (L.K.)

**Keywords:** powder coatings, organic coatings, corrosion tests, quality requirements, powder paints

## Abstract

Powder coatings are widely applied for corrosion protection of steel, aluminum, and hot dip galvanized steel in a variety of corrosive environments. Powder coatings are subjected to a number of strict laboratory tests to determine their mechanical properties, corrosion resistance, and color stability. Among European quality certificates for powder coatings applied to galvanized steel, the most commonly recognized are GSB-ST and Qualisteelcoat certificates, which also refer to the EN 13438 standard. Certificates of quality for powder coatings are constantly updated according to the latest research results and experience of specialists operating in the field of corrosion protection. This paper presents an experimental evaluation of how the required length of selected accelerated corrosion tests can affect the final assessment of powder coatings. On the example of two powder painting systems: polyester as well as based on epoxy and polyester resins, the paper presents the influence of the time of accelerated corrosion tests: ISO 6270, ISO 9227 (Neutral Salt Spray and Acetic Acid Salt Spray), and ISO 3231 on the protective properties of the coatings. The results of damage assessment according to ISO 4628 have been correlated with the requirements of particular quality specifications. Additionally, based on FTIR (Fourier Transform Infrared Spectroscopy) and EIS (Electrochemical Impedance Spectroscopy) analyses, the influence of the applied corrosion tests on the degradation degree of the coatings studied has been presented. The paper aims to present a tests for those powder coating systems applied to facilities for which the main requirement is corrosion resistance rather than aesthetics.

## 1. Introduction

Powder coatings are widely used for anti-corrosion protection of aluminum and steel elements. They may be a separate protection or together with the protective zinc coating they create a duplex system, in which the paint coating protects the zinc coating against harmful effects of atmospheric factors increasing the durability of protection [1]. An important aspect of the applied duplex coatings is also the appearance of the paint coating, which can be obtained by applying paint coatings of desired color, structure, or effect.

The powder coatings market is growing rapidly, and the share of powder coatings in the industrial coating market is steadily increasing. It is forecasted that by 2024 powder coatings in the world industrial coatings market share will exceed 13% [2]. Powder paints are more environmentally friendly than liquid paints, as they almost do not emit VOCs (volatile organic compounds) during application and curing [3]. Additionally, powder coatings not consumed during the deposition process can be reused. The limitation in the use of powder coatings is the size of the parts being protected, limited by the size of the furnaces used for curing of thermoset coatings or baths used for preparation of the surface by immersion before painting. Due to their high chemical, mechanical, and UV resistance [4], powder coatings are widely used to protect elements and construction that are operated both in external (facades, industrial buildings) and internal (equipment housings) conditions. Powder coatings can be modified to obtain specific functional properties, such as anti-graffiti, anti-bacterial, flame retardant, antistatic, abrasion resistant, or with special visual effects (metallic or wood effect).

Thermoset coatings are a broad group of materials, including different types of resins and hardeners [5]. Currently, the trends in the powder coatings market include improving their durability, lowering the curing temperature, and environmental aspects of paint manufacturing [6]. 

Powder coatings used for corrosion protection of metals, including hot dip galvanized steel, must meet a number of quality requirements for durability and corrosion resistance. The quality requirements for liquid paint systems are well known and included in the ISO 12944 series of standards [7,8,9,10,11,12,13,14,15]. However, the ISO 12944 standards are limited to coating materials that dry or cure under ambient conditions. The ISO 12944 series of standards in their scope of application clearly excludes powder coatings, stove enamels and thermosetting coating materials. The requirements for powder paints are contained in a series of standards ISO 8130 “Coating powders” [16,17,18,19,20,21,22,23,24,25,26,27,28,29,30]. 

The quality requirements for the most commonly used in Europe powder coatings are covered by the following standards or international requirements:EN 13438 Paints and varnishes—Powder organic coatings for hot dip galvanized or sherardized steel products for construction purposes [30];Qualisteelcoat requirements for organic coatings “Qualisteelcoat—International Quality Label for Coated Steel—technical specification” [31];GSB International Specification ST 663—International Quality Regulation For The Coating of Building Components [32].

Despite the exclusion of powder coatings from the ISO 12944 series of standards, the corrosivity categories according to ISO 12944-2/ISO 9223 [8,33] and the corrosion protection lifetimes according to ISO 12944-1 are well known and also used in specifications for powder coating systems (e.g., Qualisteelcoat, GSB-ST). A general classification of corrosive atmospheres according to ISO 12944-2 is given in Appendix A. The resistance of powder coating systems in a given corrosion environment depends on the substrate material, the pretreatment (presence of conversion coatings), the applied metal coating in the duplex system (single hot dip galvanizing, continuous hot dip galvanizing), the type of primer (e.g., electrophoretic coating, zinc-containing/zinc-free primer), the number and thickness of the paint coats in the corrosion protection system [34].

Each specification for the approval of powder coating systems offers its own set of accelerated corrosion tests and different test durations, often higher than the ISO 12944-6 tests for liquid paint coatings. This seems unreasonable, taking into account that the artificial ageing techniques simulate the degree of corrosivity of the atmosphere during a given lifetime according to ISO 12944-1 ÷ 2.

Table 1 presents the types and required exposure times of selected accelerated corrosion tests of organic powder coatings applied to galvanized steel according to the quality requirements of EN 13438, GSB-ST and Qualisteelcoat [35].

Industry specifications such as GSB-ST and Qualisteelcoat in their scope also have requirements for chemical surface pretreatment formulations, blasting materials, powder coating quality, and requirements for coaters. For the purpose of this publication, we focused only on comparing the effects of time and type of corrosion tests on powder coatings. The most significant differences in accelerated corrosion test times relate to the determination of moisture resistance according to ISO 6270-1/2 [36,37] and neutral salt spray (NSS) according to ISO 9227 [38]. In addition, the acid salt spray test (AASS) is only required by EN 13438 for coatings designated as Class 2. The requirements for Class 1 and Class 2 coatings do not differ with respect to the mechanical and other corrosion properties of the coating. 

In order to verify the influence of accelerated corrosion testing times required by various quality requirements on the properties of organic powder coatings, two systems were selected for testing: polyester as well as based on epoxy and polyester resins. In the present work, the effects of various corrosion factors on the failure of the tested coatings were determined as well as their barrier properties based on electrochemical impedance spectroscopy according to ISO 16773-2 standard [39].

Test panels for corrosion testing were prepared in accordance with EN 13438. Test panels were made of continuous galvanized steel, chemically pre-treated and protected with exterior powder coating systems:A polyester coating (marked P) with a specification thickness of 80 µm;Epoxy and polyester coating (marked EP + P) with a system-specific thickness of 140 µm.

A 60 µm thick epoxy coating was used as a primer coat in the two-coat systems. The topcoat in the 1- and 2-coat systems is a TGIC-free (triglycidyl isocyanurate) polyester coating. According to the requirements of Qualisteelcoat, the way of galvanizing the test panels (continuous hot-dip galvanizing) and the chemical pretreatment of the substrate before painting (phosphating) allow to achieve high durability (H) for the 1-layer system up to the maximum corrosion class C3–H and for the 2-layer system up to the corrosion class C4–H according to ISO 12944-2.

The selection of accelerated ageing tests for P and EP + P paint systems dedicated according to Qualisteelcoat for C3 and C4 corrosive environments takes into account different ageing methods and different analysis times. Thus, for the humidity test, the longest test time of 1000 h and the one-sided exposure method ISO 6270-1, recommended by EN 13438, are used. In this method, a test panel is placed over a water bath in such a way that controlled condensation occurs on one side of the panel when exposed to a saturated mixture of water vapor and air heated to T = 38 ± 2 °C. The test according to ISO 6270-2 in CH/CC method (Constant Humidity/ Constant Condensation) is carried out under similar conditions (temperature T = 40 ± 3 °C; RH ≈ 100%) and is performed in order to obtain Qualisteelcoat/GSB-ST approval. 

For both paint systems, the longest test time for resistance to neutral salt spray NSS (750 h) was applied. In addition, the test for resistance of coatings to acidic salt spray AASS (480 h) was carried out, which is applicable for EN 13438 standard for Class 2. 

The test for the resistance of coatings to humid atmospheres containing SO_2_ simulates an industrial atmosphere and is not a commonly used test even for liquid paint coatings. To determine the influence of humid atmospheres containing SO_2_ on the tested paint systems, an intermediate test time of 240 h, as required by ISO 3231 [40], was used in this study.

Although for the GSB-ST quality mark for galvanized steel, the aging tests listed in Table 1 are performed on a single coating, for comparison purposes, tests were performed on both a single-coat and a two-coat system. To summarize, in order to compare the differences in the degree of coating degradation when subjected to accelerated testing in accordance with the requirements of GSB-ST, Qualisteelcoat and EN 13438, the following ageing tests were performed on two powder coating systems, polyester as well as based on epoxy and polyester resins, applied to a hot dip galvanized steel:Determination of resistance to humidity according to ISO 6270-1 (single- sided exposure): test duration of 1000 h;Neutral salt spray test according to ISO 9227: exposure time of 750 h;Acetic acid salt spray test according to ISO 9227: exposure time of 480 h;Determination of resistance to humid atmospheres containing sulphur dioxide: exposure time of 240 h.

The corrosion testing times and methods were chosen according to EN 13438. Since the SO_2_ resistance test only considers the evaluation of the damage on non-cut samples, while the GSB-ST requirements also require the evaluation of cut samples, despite the shorter testing time—the cut samples were also evaluated. The system tested was based on powder paints, which show good corrosion properties on aluminum substrates and are subjected to preliminary corrosion tests in IBDiM (Road and Bridge Research Institute, Warsaw, Poland) laboratory. For the purposes of publication and comparative research, the information on the coatings composition and preparation is confidential.

## 2. Materials and Methods

Test panels for corrosion testing were prepared in accordance with EN 13438 standard. Furthermore, 1 mm thick plates were made of continuous galvanized steel, designated DX51D-Z275 (according to EN 10346 [41]), chemically pre-treated (zinc phosphating), and protected with exterior powder coating systems:A polyester coating (marked P) with a specification thickness of 80 µm;Epoxy and polyester coating (marked EP + P) with a system-specific thickness of 140 µm.

Powder coatings thickness was determined according to ISO 2808 method 7B.2 [42] using Elcometer 456 electromagnetic measuring device (Elcometer, Manchester, United Kingdom). The instrument indications were verified with coating thickness standards RJ02783 (23.6 µm) and KC3990 (246.5 µm). The thickness of the powder coatings was determined by taking 10 measurements on each test panel.

Powder paint coatings used to protect galvanized test panels were characterized using a Nicolet iS 10 Thermo Scientific FTIR spectrometer (Thermo Fisfer Scientific Inc., Waltham, USA) in the wave number range of 4000–400 cm^−1^. The spectra of paint coatings were acquired according to EN 1767 [43] using the internal reflection-based method (ATR) (Attenuated Total Reflectance) at a resolution of 4 cm^−1^. The crystal used was diamond. The expanded uncertainty of peak determination (for 95% confidence level and expansion factor k = 2) is 1.4 cm^−1^.

The adhesion of powder coatings before and after accelerated corrosion tests was tested using the cut grid method in accordance with ISO 2409 [44].

Testing of the resistance to continuous condensation at single-sided exposure was carried out according to ISO 6270-1 in the Braive humidity chamber. Water vapor condensation occurs on the coating surfaces tested at an angle of 60 ± 5° by exposing the painted side of the plate to a saturated mixture of air and water vapor heated to 38 ± 2 °C, while the back side is exposed to air at a temperature of 23 ± 2 °C. The total test time for the powder coatings was 1000 h (according to EN 13438), while the test samples were subjected to intermediate evaluations after: 240 h, 480 h and 720 h. 

The neutral salt spray (NSS) corrosion test was carried out according to the ISO 9227 standard in a Kohler HKT 500 chamber ( Köhler Automobiltechnik, Lippstadt, Germany). The test was carried out at a temperature of 35 ± 2 °C in a salt spray environment with a concentration of 50 ± 5 g/L, which continuously moistens the surface of the tested coatings. The pH value of the sprayed solution is within the range of 6.5–7.2. In accordance with EN 13438, powder coatings are tested over a period of 750 h with intermediate evaluation after 480 and 720 h. 

Corrosion test in acetic acid salt spray (AASS) was carried out according to ISO 9227 in the Q-FOG CCT-600 chamber (country-Lab Corporation, Westlake, USA). Composition of solution sprayed on tested coatings: NaCl 50 ± 5 g/L with the addition of 1% CH_3_COOH, pH of the solution: 3.1–3.3. Temperature in the chamber was 35 ± 2 °C. According to EN 13438, powder coatings under given conditions are tested for 480 h with intermediate evaluation after 120 h.

Kesternich test was carried out according to ISO 3231 in Ascott 2409 chamber (Ascott Analytical Equipment Limited, Tamworth, United Kingdom) using 0.2 L of SO_2_. 10 test cycles were carried out, each cycle consisted of a step of heating an hermetic chamber containing 2 ± 0.2 L of water in the presence of SO_2_ to the temperature of 40 ± 3 °C, maintaining this temperature for a period of 8h and relative humidity of RH = 100% and transferring the plates to an air conditioning chamber for a period of 16 h. After each cycle, the water is changed and 0, l L of SO_2_ are introduced into the test chamber to repeat the 24 h test cycle. 

Panels without scribes as well as with scribe according to ISO 17872 [45] were subjected to NSS/AASS and ISO 3231 tests (resistance to humid atmosphere containing sulphur dioxide). 

The uncut samples were evaluated according to ISO 4628-2 ÷ 5 [46,47,48,49] for the presence of blistering, rusting, cracking and flaking of the coating. The method of evaluating coating damage according to ISO 4628-2 ÷ 5 and the damage designations established in this series of standards are shown in Appendix A. The cut samples were additionally evaluated for the degree of delamination and corrosion around the scribe according to ISO 4628-8 [50].

The impedance of the powder coatings before and after the corrosion tests was determined using the Iviumstat measuring set according to the ISO 16773-2 standard. The measurements were performed in a three-electrode configuration, in a 3.5% sodium chloride solution, within a frequency range of 10^5^ Hz to 10^−1^ Hz. The sinusoidal amplitude of 10 mV was applied. The working electrode was the test sample, the counter electrode was a platinum grid, and the reference electrode was a calomel electrode. After each aging test of the coatings was completed, impedance spectra were collected and equivalent circuits were selected to match the recorded spectra. The impedance spectra were analyzed using the Zview 4 fitting procedure (Scribner Associates Inc., Southern Pines, USA). The results are presented as the logarithm of the impedance modulus as a function of the frequency. The value of the logarithm of the impedance modulus at a frequency of 0.1 Hz allows to determine the protective properties of the paint system, i.e., the higher the value of the impedance modulus the higher the barrier properties of the protection.

## 3. Results and Discussion

### 3.1. Analysis of FTIR Spectra

Figure 1 presents the FTIR spectra for the polyester powder coatings from 600 to 4000 cm^−1^, after accelerated corrosion testing. Clear peaks from the functional groups of the polyester resin are observed on the FTIR absorbance spectrum: a carbonyl stretching vibration (C=O) at 1716 cm^−1^, C–O stretching vibration bands in the region 1100–1300 cm^−1^, with the highest absorption peak at 1243 cm^−1^ [51,52]. On the FTIR spectrum also the peaks assigned to asymmetric and symmetric C-H stretching vibrations of CH_3_ and CH_2_ groups (in the 3000–2800 cm^−1^ region) as well as peaks attributed to C–CH_2_–C and C–CH_3_ bending vibrations (at 1474 and 1373 cm^−1^). [51,52]. The bands at 726 cm^−1^ and 874 cm^−1^ are typical for C–H rocking vibrations of the methyl groups and CaCO_3_ respectively, which originated from the crosslinker and the filler.

From the comparative analysis of FTIR spectra, it can be observed that the applied accelerated aging test times do not cause degradation of the polymer bonds. The thermoset polyester coatings exhibit good resistance to the applied corrosion factors such as: humidity, chloride-rich atmosphere, acid salt spray, or humid atmospheres rich in SO_2_, which is related to the curing mechanism [52].

### 3.2. Measuring the Thickness and Adhesion of Powder Coatings on Test Panels

The thickness of the powder coating systems used is consistent with the specific thickness of the systems recommended by Qualisteelcoat and GSB-ST. Measurements were conducted on all test panels subjected to corrosion testing. The thicknesses shown in Table 2 are average values, and each of the test panels meets the coating thickness requirements for C3 (P: polyester) and C4 (EP + P: epoxy + polyester) environments. The adhesion of the coatings before corrosion testing is correct and determined as grade 0 according to ISO 2808.

### 3.3. Evaluation of Powder Coating Damage on Panels without a Scribe

Table 3 shows the results of damage assessment according to ISO 4628-2 ÷ 5 of powder coatings on uncut panels after selected accelerated corrosion tests. From the obtained test results, it can be observed that the investigated polyester (P) system is not resistant to humidity at 1000 h. Based on the intermediate evaluation of the samples, after test times of 240 h and 480 h, it was found that no blistering was observed for the polyester coating even after the 480 h required for a two-coat system for C4–H environment by Qualisteelcoat. The appearance of the polyester coating after the NSS test at 750 h is shown in Figure 2. All other accelerated corrosion tests, i.e., resistance to neutral and acidic salt spray as well as humid atmosphere containing SO_2_, in given test times did not cause damage according to ISO 4628-2 ÷ 5 to any of the tested paint systems.

### 3.4. Evaluation of Powder Coating Damage on Panels with a Scribe

The evaluation of powder coatings in the area of artificial damage in the form of a scribe allows to determine the degree of coating delamination and to verify whether the progressive corrosion does not cause corrosion under the coating. The results of damage assessment around the scribe according to ISO 4628-2 ÷ 5 standards, after selected accelerated corrosion tests, are presented in Table 4. After 750 h of exposure, the polyester system (P) shows significant blistering. However, blistering does not appear until the 480 h test time required by Qualisteelcoat and GSB-ST. The appearance of the polyester coating near the scribe after the NSS test at 750 h is shown in Figure 3.

### 3.5. Assessment of the Soating Delamination and Corrosion around the Scribe

Table 5 and Table 6 show the results of delamination and corrosion measurements performed around the scribe, determined according to ISO 4628-8. Apart from the list of damage measurement results, the table contains acceptable values according to various quality requirements (along with recommended testing time, if it is different from the applied one). If the test is not required by a given recommendation, or if an assessment of delamination or corrosion is not required—this information is also included in the table. For example: EN 13438 requires testing to ISO 3231, but does not consider testing of artificially damaged panels. Only the color change as well as degree of blistering and corrosion are assessed. In accordance to the GSB-ST requirements, the test duration is longer (720 h); however, only the delamination of the coating around the scribe is assessed. 

Based on the measurements obtained, it can be observed that the highest degree of damage around the scribe, both in terms of coating delamination and corrosion is observed after 750 h of the NSS test. The average delamination of the polyester coating around the scribe is in the range of 3.55–5.17 mm, which makes the system fail to meet the requirements of EN 13438 for resistance to neutral salt spray. Since the average coating delamination slightly exceeds the 5 mm value, it can be presumed that after the 480 h required for GSB-ST specification, the system’s degree of coating delamination would be acceptable by these requirements. The polyester single-coat system would also meet the Qualisteelcoat requirements for acceptable coating delamination, which is 8 mm for galvanized substrates.

Determination of the degree of corrosion around a crack is not always required, despite the requirement for testing, e.g., in the case of NSS testing (GSB-ST specification) or testing in an atmosphere containing SO_2_ according to ISO 3231 (EN 13438 standard, GSB-ST specification).

For the neutral salt spray powder coating resistance test, the degree of corrosion of the substrate around the scribe would be acceptable according to the requirements of EN 13438, but would not meet the Qualisteelcoat requirements for a two-coat system (EP + P) to the C4 environment. Assuming a corrosion progression proportional over the test time, the corrosion degree would also be too high for the single-coating system (P) after a time of 480 h. The acid salt spray test showed a resistance of the tested systems to the applied test time of 480 h according to the requirements of EN 13438. In the case of the test for resistance to humid atmospheres containing SO_2_, none of the quality requirements (EN 13438, GSB-ST), require an evaluation of corrosion around the crack.

As mentioned previously, a test of coating resistance to humid atmospheres containing SO_2_ is not required at all (Qualisteelcoat), is required for a time of 240 h for panels without a scribe (EN 13438) or is required for a time of 720 h, and only the degree of coating delamination is evaluated for cut samples (GSB-ST). After a test time of 240 h, a slight delamination of the coating around the scribe was observed, averaging 0.19–0.31 mm for the P system and averaging 0–0.29 mm for the EP+P system. By considering the GSB-ST criteria and assuming a proportional progression of coating adhesion loss and corrosion around the scribe over the test time, it can be presumed that the single-coating system (P) tested would have a chance of passing the test, as two of the three test panels tested would not achieve a coating delamination exceeding 1 mm.

### 3.6. Evaluation of Powder Coatings Adhesion after Accelerated Corrosion Tests

Although coating adhesion testing after corrosion testing is not always required by the quality requirements discussed, Table 7 shows the degree of adhesion of the tested powder coatings after the applied corrosion tests presented in the Experimental Procedure section. Both the polyester and epoxy + polyester coating systems exhibited correct adhesion of the paint system to the substrate, although there was slight blistering of the polyester coating after the ISO 6270 humidity test for 1000 h.

### 3.7. Electrochemical Impedance Spectroscopy Analysis

For liquid paint coatings and according to ISO 16773 the measurements are typically obtained over the range of 10^5^ Hz to 10^−2^ Hz and the impedance modulus value |Z| is reported at 0.1 Hz. Although none of the described quality requirements (EN 13438, Qualisteelcoat, GSB-ST) for powder coatings on galvanized steel require electrochemical impedance spectroscopy (EIS) before or after corrosion tests, EIS is a quantitative method that can be used to assess barrier properties of the non-conductive organic coatings on metal surfaces [53]. The main role of EIS in organic coating characterization is to provide insight into about the properties of the protective system, such barrier properties, coating defects and the development of subcoating corrosion.

After all accelerated corrosion tests, electrochemical impedance spectroscopy analysis was performed. Figure 4, Figure 5, Figure 6 and Figure 7 show Bode plots of the impedance modulus: |Z|(f) and phase angle: θ(f) for the tested P and EP+P powder systems after accelerated corrosion testing.

From the Bode impedance modulus plots for the polyester single coating system, it can be observed that the greatest effect on the reduction of the coating’s barrier properties is due to humidity exposure at 1000 h and NSS neutral salt spray exposure at 750 h. The least effect on the impedance level of the polyester coating is observed when tested in a humid atmosphere containing SO_2_. These results correlate with the evaluation of coating damage according to ISO 4628-2 ÷ 5 and with the assessment of damage according to ISO 4628-8 (coating delamination and corrosion) around the artificial damage. For the EP + P two-coating system, the greatest effect on the reduction of the coating barrier properties is observed after exposure to a humid atmosphere containing SO_2_. 

The impedance results of the analyzed powder coatings before and after the corrosion tests are shown in Table 8. Values of the logarithm of impedance at 0.1 Hz below 6 indicate low barrier properties of the anti-corrosion system [54]. The presented measurement results clearly confirm the degrading effect of humidity at 1000 h on the polyester coating. Despite the blistering of the polyester coating around the scribe after exposure to neutral salt spray for 750 h, this type of damage was not observed on the panels without scribe, and the impedance measurements confirm the good barrier properties of this coating, as long as it remains unbroken. The analyzed two-coat powder system retains its protective properties at all given corrosion test times.

The experimental EIS data were fitted to Randles equivalent circuit modified with a constant phase element (Figure 8). Applied equivalent circuit consist of:The resistance of electrolyte solution R_s_;The constant phase element capacitance C_CPE_;The resistance of coating R_ct_.

Equivalent circuit fitted to experimental data were used to determine the R_s_, C_CPE,_ and R_ct_ values of the powder coating systems tested. The results obtained are shown in Table 9 and Table 10.

Based on the EIS results shown in Table 9 and Table 10, it can be observed that the determined value of Re (electrolyte resistance) is between 1 and 2 kΩ for all cases studied. Constant phase element (CPE) consider the non-ideal behavior of an organic coating due to the surface inhomogeneity or roughness, electrode porosity, geometric irregularities, and others. The C_CPE_ values for the polyester coating in the initial condition is 8.71 × 10^−^^12^ F, and after accelerated corrosion tests, it does not change significantly, except for the coating after the neutral salt spray resistance test at 750 h, where the determined value of the capacitance is 9. 91 × 10^−^^11^ F. The parameter n for the polyester coating describing the CPE is between 0.86 and 0.97, except for the coating after the humidity resistance test at 1000 h, where n is 0.25. The value of n is between 0 and 1, for n = 1 the CPE is considered an ideal capacitor. The charge transfer resistance between the solution and the coating Rct for the polyester coating in the initial state is 2.27 × 10^−^^11^ Ω and does not significantly decrease after accelerated corrosion tests, except for the humidity resistance test according to ISO 6270-1. In this case, the decrease in the Rct value and as well as in the n number are related to the decrease in the barrier properties of the polyester coating tested for 1000 h in the humidity chamber which was observed in the form of blistering of the coating and the occurrence of corrosion processes. 

For the epoxy + polyester two-coating system, the C_CPE_ value is 2.09 × 10^−^^10^ F in the initial state and does not change significantly after accelerated corrosion testing. The parameter n describing the CPE for the epoxy-polyester system ranges from 0.82 for the coating in the initial state to values close to 0.9, except for the coating after the test of resistance to humid atmospheres containing SO_2_ for 240 h, where n is 0.74. For the coating after the ISO 3231 test, a decrease in the phase angle value in the low frequency range was also observed, while the other accelerated corrosion tests did not change the nature of the θ(f) curve across the frequency range. 

## 4. Summary and Conclusions

The paper presents the results of accelerated aging tests of selected powder coatings performed according to different quality standards: GSB-ST 663-4, Qualisteelcoat, and EN 13438 as well as an experimental evaluation of how the required length of selected accelerated corrosion tests can affect the final assessment of powder coatings. A summary of the tests results and their evaluation to the criteria of the discussed quality requirements is presented in Table 11.

The differences in the evaluation criteria and required accelerated corrosion test durations are so significant that the tested two-coat epoxy + polyester (EP+P) coating system applied to continuously galvanized steel, despite its good corrosion resistance confirmed with impedance level, would only have a chance of achieving a positive evaluation according to the quality requirements of EN 13438 (assuming it passed the other mechanical and weathering tests).

Different quality requirements for powder coatings presented in the paper raise the question of how they relate to the real corrosivity of the environment that affects the protections of steel structures systematized in ISO 12944-2 in the form of a classification into corrosivity classes of the environment. Among the quality specifications for powder coatings, only the Qualisteelcoat requirements relate the corrosion test durations (resistance to humidity and neutral salt spray) to the environmental corrosivity classes and they are equivalent to the test durations for liquid paint coatings specified in ISO 12944-6, which seems to be a fully justified approach.

As can be observed in Table 1, some of the required corrosion test durations, especially the powder coating humidity resistance test time contained in the GSB-ST and EN 13438 requirements are more restrictive than those specified in ISO 12944-6 for liquid paint coatings. Moreover, despite the same NSS test durations for liquid paint coatings according to ISO 12944-6 and powder coatings according to Qualisteelcoat, the criterium for evaluating corrosion around the scribe is more restrictive for powder coatings. The acceptable corrosion around scribe is 1 mm compared to 1.5 mm specified in ISO 12944-6 standard for liquid paints. Some of the quality specifications presented also require additional tests, such as a resistance test to humid atmospheres containing SO_2_ or acetic acid salt spray (AASS), which might be related to the low thickness of powder coatings relative to liquid paint systems and thus possible lower barrier capabilities to acidic environments, however, it was not confirmed in our study. None of the powder systems tested was excluded by negative results in these accelerated corrosion tests including tests which are not covered by ISO 12944-6 like resistance to humid atmosphere containing SO_2_ and acidic salt spray.

Based on the obtained results, we propose to bring the requirements for anti-corrosion systems of powder paints used in municipal and road infrastructure facilities closer to the requirements for anti-corrosion systems of liquid paints. For this application also radiation resistance according to ISO 16474-3 should be enough, leaving Florida test for architectural solutions. But mainly the most demanding neutral salt spray test should be used, that is 750 h as required by the standard EN 13438. This will eliminate systems that do not provide adequate corrosion resistance. This problem is now very common on noise barriers where powder systems corrode after a few years. Such requirements will allow for the creation of a group of powder coating systems adapted to the requirements of city and road infrastructure, which will be able to obtain a certificate in a shorter time and for less money.

Certificates of quality for powder paint products are provided by associations that have been trusted by paint manufacturers, investors, contractors and quality inspectors for many years. Since European quality recommendations for powder coatings are based on International Organization for Standardization and European Standards, they are constantly updated according to the latest knowledge, research results and experience of specialists operating in the field of anti-corrosion protection. However, it must be considered that behind each powder coating quality certificate, different quality requirements are specified by a given standard. This paper presents a practical example of how differences in the required length of selected accelerated corrosion tests can affect the final assessment of powder coatings.

## Figures and Tables

**Figure 1 materials-14-06547-f001:**
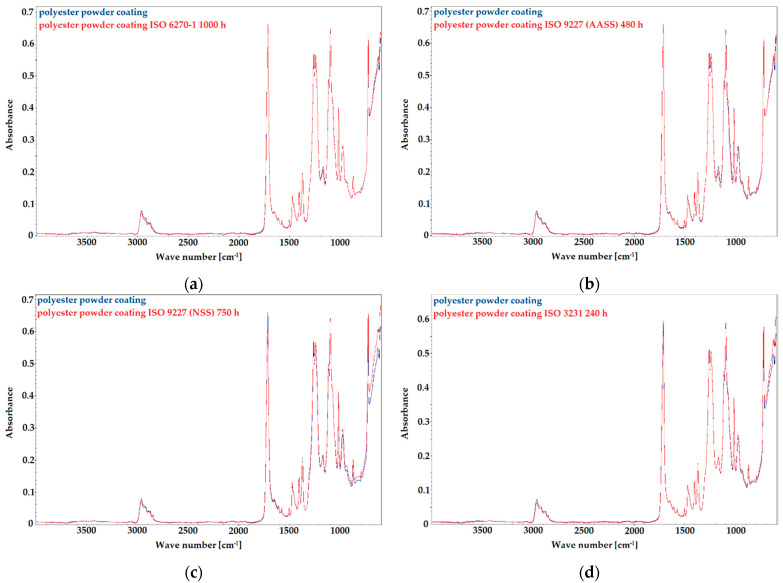
FTIR spectra of polyester topcoat after accelerated corrosion tests according to: (**a**) ISO 6270-1 for 1000 h; (**b**) ISO 9227 AASS for 480 h; (**c**) ISO 9227 NSS for 750 h; (**d**) ISO 3231 for 240 h.

**Figure 2 materials-14-06547-f002:**
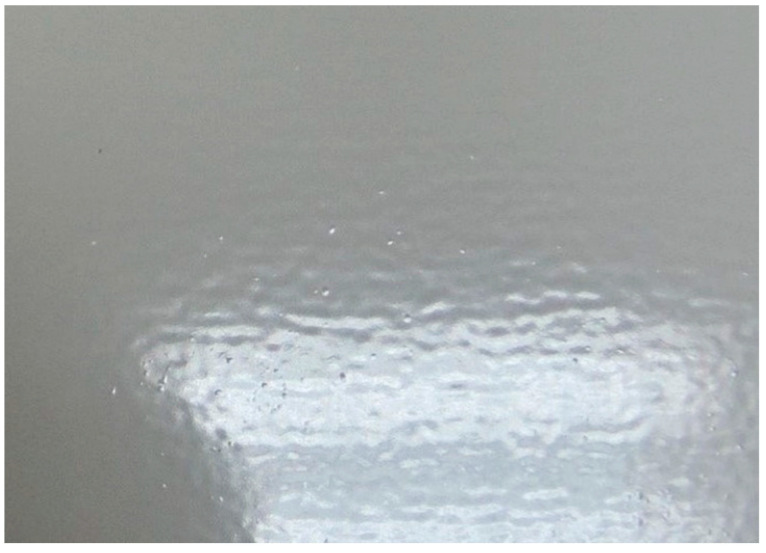
Blistering of the polyester coating after 1000 h humidity resistance test.

**Figure 3 materials-14-06547-f003:**
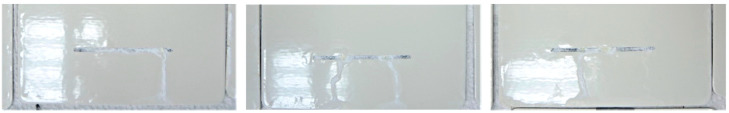
The appearance of the polyester coating near the scribe after the NSS test at 750 h.

**Figure 4 materials-14-06547-f004:**
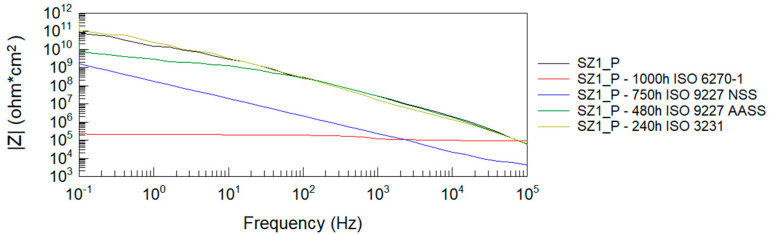
Bode plots of polyester coating before and after accelerated corrosion tests: impedance modulus |Z|(f).

**Figure 5 materials-14-06547-f005:**
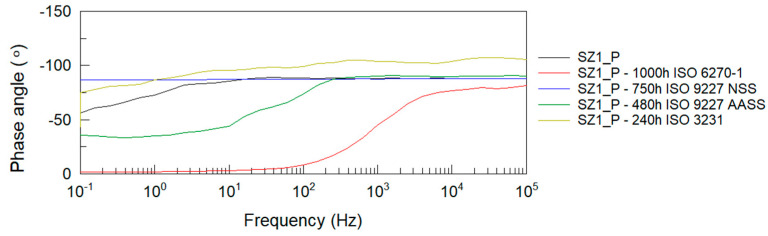
Bode plots of polyester coating before and after accelerated corrosion tests: phase angle θ(f).

**Figure 6 materials-14-06547-f006:**
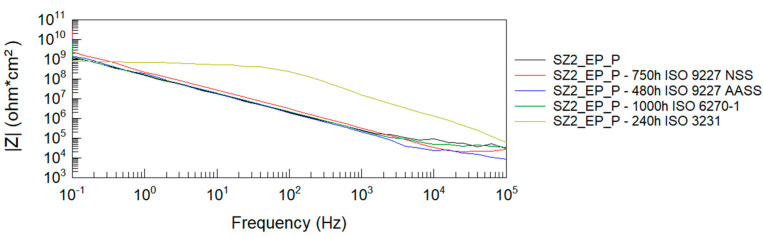
Bode plots of epoxy + polyester coating system before and after accelerated corrosion tests: impedance modulus |Z|(f).

**Figure 7 materials-14-06547-f007:**
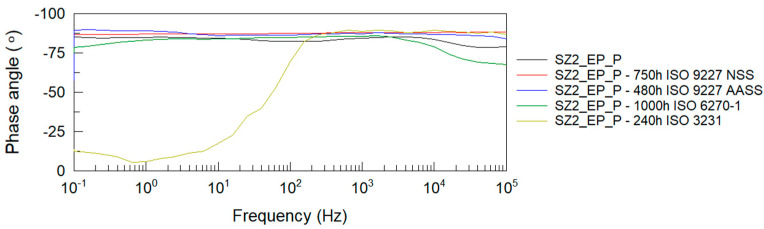
Bode plots of epoxy + polyester coating system before and after accelerated corrosion tests: phase angle θ(f).

**Figure 8 materials-14-06547-f008:**
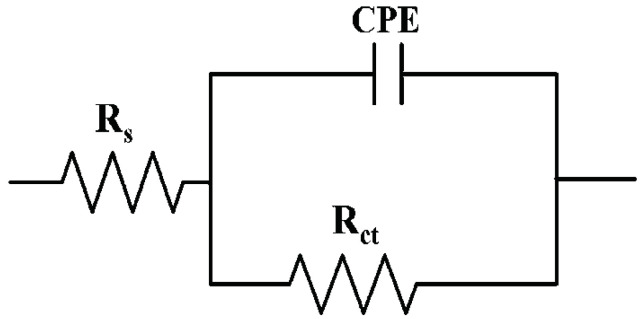
Schematic of the Randles equivalent circuit modified with a constant phase element.

**Table 1 materials-14-06547-t001:** Required Accelerated Ageing Test Times according to Different Powder Coating Quality Assessment Procedures.

Characteristic Studied		Required Exposure Time		
		EN 13438	Qualisteelcoat	GSB-ST 663-4/7
**Radiation resistance using laboratory light source**	**ISO 16474-3**	-	-	1000 h (class “Florida 10”)
				600 h (class “Florida 3&5”)
				300 h (class “Florida 1”)
	**ISO 16474-2**	1000 h	1000 h ^1^	-
**Resistance to humidity**	**ISO 6270-1**	1000 h	-	-
	**ISO 6270-2**	-	120 h C2-H	1000 h
			240 h C3-H	
			480 h C4-H	
			720 h C5-H	
**Salt spray resistance**	**ISO 9227-NSS**	750 h (Class 1)	240 h C2-H	480 h
			480 h C3-H	
			720 h C4-H	
			1440 h C5-H ^2^	
	**ISO 9227-AASS**	480 h (Class 2)	-	-
**Resistance to humid** **atmospheres containing SO_2_**	**ISO 3231**	240 h	-	720 h
**Natural weathering test (Florida test)**	**ISO 2810**	1 year	1 year ^3^	10 years (class “Florida 10”)
				3/5 years (class “Florida 3&5”)
				1 year (class “Florida 1”)

^1^ For corrosivity classes above C3–H.; ^2^ Alternate climate test for 1680 h acc. to ISO 12944-6 annex B.; ^3^ Coating systems for outdoor applications.

**Table 2 materials-14-06547-t002:** Average thickness of the P and EP+P coating systems and coatings adhesion grade determined in accordance with the ISO 2409 standard.

Coating System	Average Thickness of the Coating System [μm]	Coating Adhesion Grade According to ISO 2409		
		1	2	3
**P**	91.5 ± 5.6 ^1^	0	0	0
EP + P	158.3 ± 8.6 ^1^	0	0	0

^1^ Expanded standard uncertainty k = 2; 95% confidence level.

**Table 3 materials-14-06547-t003:** Results of the Damage Assessment of Uncut Testing Panels after Selected Accelerated Corrosion Tests.

Accelerated Corrosion Test	Coating System	Damage Assessment According to ISO 4628-2 ÷ 5											
		Blistering			Rusting			Cracking			Flaking		
		1	2	3	1	2	3	1	2	3	1	2	3
ISO 6270 (1000 h )	P	3(S2)	2(S2)	3(S2)	Ri0	Ri0	Ri0	0(S0)	0(S0)	0(S0)	0(S0)	0(S0)	0(S0)
	EP + P	0(S0)	0(S0)	0(S0)	Ri0	Ri0	Ri0	0(S0)	0(S0)	0(S0)	0(S0)	0(S0)	0(S0)
ISO 9227—NSS (750 h)	P	0(S0)	0(S0)	0(S0)	Ri0	Ri0	Ri0	0(S0)	0(S0)	0(S0)	0(S0)	0(S0)	0(S0)
	EP + P	0(S0)	0(S0)	0(S0)	Ri0	Ri0	Ri0	0(S0)	0(S0)	0(S0)	0(S0)	0(S0)	0(S0)
ISO 9227—AASS (480 h)	P	0(S0)	0(S0)	0(S0)	Ri0	Ri0	Ri0	0(S0)	0(S0)	0(S0)	0(S0)	0(S0)	0(S0)
	EP + P	0(S0)	0(S0)	0(S0)	Ri0	Ri0	Ri0	0(S0)	0(S0)	0(S0)	0(S0)	0(S0)	0(S0)
ISO 3231 (240 h)	P	0(S0)	0(S0)	0(S0)	Ri0	Ri0	Ri0	0(S0)	0(S0)	0(S0)	0(S0)	0(S0)	0(S0)
	EP + P	0(S0)	0(S0)	0(S0)	Ri0	Ri0	Ri0	0(S0)	0(S0)	0(S0)	0(S0)	0(S0)	0(S0)

**Table 4 materials-14-06547-t004:** Results of the Damage Assessment near the Scribe after Selected Accelerated Corrosion Tests.

Accelerated Corrosion Test	Coating System	Damage Assessment near the Scribe According to ISO 4628-2 ÷ 5											
		Blistering			Rusting			Cracking			Flaking		
		1	2	3	1	2	3	1	2	3	1	2	3
**ISO 9227—NSS (750 h)**	P	3(S4)	2(S5)	5(S5)	Ri0	Ri0	Ri0	0(S0)	0(S0)	0(S0)	0(S0)	0(S0)	0(S0)
	EP + P	0(S0)	0(S0)	0(S0)	Ri0	Ri0	Ri0	0(S0)	0(S0)	0(S0)	0(S0)	0(S0)	0(S0)
ISO 9227—AASS (480 h)	P	0(S0)	0(S0)	0(S0)	Ri0	Ri0	Ri0	0(S0)	0(S0)	0(S0)	0(S0)	0(S0)	0(S0)
	EP + P	0(S0)	0(S0)	0(S0)	Ri0	Ri0	Ri0	0(S0)	0(S0)	0(S0)	0(S0)	0(S0)	0(S0)
ISO 3231 (240 h)	P	0(S0)	0(S0)	0(S0)	Ri0	Ri0	Ri0	0(S0)	0(S0)	0(S0)	0(S0)	0(S0)	0(S0)
	EP + P	0(S0)	0(S0)	0(S0)	Ri0	Ri0	Ri0	0(S0)	0(S0)	0(S0)	0(S0)	0(S0)	0(S0)

**Table 5 materials-14-06547-t005:** The Delamination Width of the Coating around the Scribe after Selected Accelerated Corrosion Tests and the Aceptable Width of the Coating Delamination according to the Different Quality Requirements.

Accelerated Corrosion Test	Coating System	Measurement of the Damage around an Artificial Defect According to ISO 4628-8			Quality Requirements		
		Delamination around the Scribe [mm]			EN 13438	Qualisteelcoat	GSB-ST
		1	2	3			
ISO 9227—NSS (750 h)	P	3.55 ± 0.61	5.17 ± 0.75	5.13 ± 0.76	≤5 mm	≤8 mm (480 h)	≤5 mm (480 h)
	EP + P	1.83 ± 0.54	1.54 ± 0.38	1.07 ± 0.36		≤8 mm (720 h)	not applicable
ISO 9227—AASS (480 h)	P	1.46 ± 0.37	2.45 ± 0.77	3.60 ± 0.96	≤5 mm	not applicable	not applicable
	EP + P	0.33 ± 0.16	0.39 ± 0.19	0.34 ± 0.12			
ISO 3231 (240 h)	P	0.31 ± 0.12	0.24 ± 0.11	0.19 ± 0.08	no evaluation required	not applicable	≤1 mm (720 h)
	EP + P	0.29 ± 0.07	0	0			not applicable

**Table 6 materials-14-06547-t006:** The Width of Substrate Corrosion around the Scribe after Selected Accelerated Corrosion Tests and the Acceptable Width of the Substrate Corrosion according to the Different Quality Requirements.

Accelerated Corrosion Test	Coating System	Measurement of the Damage around an Artificial Defect According to ISO 4628-8			Quality Requirements		
		Corrosion around the Scribe [mm]			EN 13438	Qualisteelcoat	GSB-ST
		1	2	3			
ISO 9227—NSS (750 h)	P	2.08 ± 0.95	2.03 ± 0.74	2.43 ± 1.12	≤5 mm	≤1 mm (480 h)	no evaluation required
	EP + P	1.08 ± 0.72	1.28 ± 0.73	1.21 ± 0.36		≤1 mm (720 h)	not applicable
ISO 9227—AASS (480 h)	P	0.61 ± 0.17	0.84 ± 0.21	1.02 ± 0.20	≤5 mm	not applicable	not applicable
	EP + P	0.19 ± 0.08	0.28 ± 0.04	0.23 ± 0.16			
ISO 3231 (240 h)	P	0.23 ± 0.12	0.18 ± 0.08	0.19 ± 0.07	no evaluation required	not applicable	no evaluation required
	EP + P	0.32 ± 0.11	0.14 ± 0.09	0.16 ± 0.08			not applicable

**Table 7 materials-14-06547-t007:** Coatings Adhesion Grade Determined in Accordance with the ISO 2409 Standard after Selected Accelerated Corrosion Tests and Acceptable Adhesion according to the Different Quality Requirements.

Accelerated Corrosion Test	Coating System	Coating Adhesion Grade According to ISO 2409			Quality Requirements		
					EN 13438	Qualisteelcoat	GSB-ST
		1	2	3			
ISO 6270 (1000 h)	P	0	0	0	0	no evaluation required	no evaluation required
	EP + P	0	0	0	0		not applicable
ISO 9227—NSS (750 h)	P	0	0	0	no evaluation required	0/1	no evaluation required
	EP + P	0	1	0		0/1	not applicable
ISO 9227—AASS (480 h)	P	1	0	0	no evaluation required	not applicable	no evaluation required
	EP + P	0	0	0			not applicable
ISO 3231 (240 h)	P	0	0	0	no evaluation required	not applicable	no evaluation required
	EP + P	1	0	0			not applicable

**Table 8 materials-14-06547-t008:** Impedance Values of the Investigated Powder Coatings at 0.1 Hz.

	Coating System	Before Tests	ISO 6270-1 (1000 h)	ISO 9227-NSS (750 h)	ISO 9227-AASS (480 h)	ISO 3231 (240 h)
**Impedance values Z for 0,1 Hz [Ωcm^2^]**	P	7.96 × 10^10^	2.18 × 10^5^	1.66 × 10^9^	7.86 × 10^9^	1.04 × 10^11^
	EP + P	1.17 × 10^9^	1.37 × 10^9^	2.42 × 10^9^	1.48 × 10^9^	8.26 × 10^8^
**Phase angle values** **θ for 0,1 Hz [°]**	P	−56.1	–1.9	–86.7	–35.8	–75.1
	EP + P	–85.3	–78.4	–86.6	–86.6	–12.5

**Table 9 materials-14-06547-t009:** EIS Results of Polyester Powder Coating before and after Accelerated Corrosion tests.

Polyester Coating	R_s_ (Ω)	R_ct_ (Ω)	C_CPE_ (F)	n
**Initial state**	1189	2.27 × 10^11^	8.71 × 10^−12^	0.86
**ISO 6270-1 (1000 h)**	1145	1.25 × 10^8^	8.50 × 10^−12^	0.25
**ISO 9227-NSS (750 h)**	1147	8.17 × 10^11^	9.91 × 10^−11^	0.95
**ISO 9227-AASS (480 h)**	1200	3.06 × 10^12^	3.75 × 10^−12^	0.91
**ISO 3231 (240 h)**	1013	4.31 × 10^12^	5.06 × 10^−12^	0.97

**Table 10 materials-14-06547-t010:** EIS Results of Epoxy + Polyester Powder Coating System before and after Accelerated Corrosion Tests.

Epoxy + Polyester Coating System	R_s_ (Ω)	R_ct_ (Ω)	C_CPE_ (F)	n
**Initial state**	1115	1.03 × 10^13^	2.09 × 10^−10^	0.82
**ISO 6270-1 (1000 h)**	1192	5.04 × 10^11^	1.46 × 10^−10^	0.89
**ISO 9227—NSS (750 h)**	1184	2.07 × 10^13^	3.38 × 10^−11^	0.99
**ISO 9227—AASS (480 h)**	1154	7.78 × 10^11^	8.33 × 10^−11^	0.98
**ISO 3231 (240 h)**	1114	1.86 × 10^12^	4.56 × 10^−10^	0.74

**Table 11 materials-14-06547-t011:** Summary of the Accelerated Corrosion Test Results.

IS THE SYSTEM COMPLIANT WITH THE QUALITY REQUIREMENTS				
Accelerated Corrosion Test	Coating System	Quality Requirements		
		EN 13438	Qualisteelcoat	GSB-ST
ISO 6270	P	NOT (1000 h)	YES (240 h)	NOT (1000 h)
	EP + P	YES (1000 h)	YES (480 h)	not applicable
ISO 9227-NSS	P	NOT (750 h)	NOT (480 h) ^1^	YES (480 h) ^1^
	EP + P	YES (750 h)	NOT (720 h)	not applicable
ISO 9227-AASS	P	YES (480 h)	not applicable	not applicable
	EP + P	YES (480 h)		not applicable
ISO 3231	P	YES (240 h)	not applicable	not performed ^2^
	EP + P	YES (240 h)		not applicable

^1^ Assuming a progression of coating adhesion loss and corrosion around the scribe to be proportional over the test period; ^2^ Intermediate test time according to ISO 3231 was applied, see Section 2.

## Data Availability

The results of the study are not placed in any publicly archived datasets.

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
