# Peer review of "Accelerated Corrosion Tests in Quality Labels for Powder Coatings on Galvanized Steel—Comparison of Requirements and Experimental Evaluation"

_materials, 2021, doi:10.3390/ma14216547_

Round 1
Reviewer 1 Report
The authors investigated the performance of two coatings under different test conditions which might be helpful to update the test standards for coatings. It can be accepted after revision.
(1) Please retake pictures for Fig.2 and 3 to avoid light reflection from the sample.
(2) For Fig.3, it is better to provide the macroscopy of polyester coating after tests for different times for comparison.
(3) In table 2, it seems that both coatings become thicker after the test, about 10 um. Please explain this result.
(4) Please provide EIS along with equivalent circuits besides the Bode.
Author Response
Dear Reviewer #1
We thank you for your valuable comments that improved the quality of the publication. We have attached images to our responses, so please download the attached file.

Reviewer 2 Report
This manuscript can be interesting to a readership involved in coatings evaluation. Several standards are cited, applied and results are reported. However in few cases a direct reference to the classification coding of the standards is not helpful to readers that are not familiar. For example reference to Coating grade according to ISO 2409 (Table 2) or class abbreviations in other Tables (i.e. Table 3) makes no sense if it is not defined or explained in a Supplementary Material.
The overall concept of combining testing protocols, excluding/avoiding some and in general justify a faster and less expansive quality certification is a subject of interest. Especially when coatings prepared by different technologies can be used in similar or exactly the same applications or for similar purposes.
L19 please do not use abbreviations with no prior definition of meaning i.e. NSS and AASS
L23 authors claim that the paper aims to introduce test for powder coating systems that are to be applied in corrosive conditions. How do the authors do that? Where in the text? Authors suggest to avoid the humid atmosphere containing SO2 and/or acidic salt spray tests. Please explain / justify.
L23 typo error exists
L406 Authors suggest to “withdraw the test requirements like resistance to humid atmosphere containing SO2 and acidic salt spray“ for surfaces exposed to urban or road areas. How is this justified considering the SOx and NOx emissions by fossil fuel based combustion and burner units that exist? Thus the pH of precipitation is very often acidic. The authors need to provide more evidence that a salt spray test can substitute the chemical corrosion induced by the SOx or acid presence. I am not sure that this is feasible in the present manuscript thus I would suggest to re-write this part.
L411-414 Authors claim that limited thickness of coatings is responsible for their low corrosion resistance. I believe this is a rather overismplistic approach as corrosion resistance is not just due to barrier properties.
Several aging tests according to different quality standards have been performed in 2 types of coatings authors prepared. Results are nicely collected and presented and the overall technical features of the manuscript are good. However as a reader I do not get some solid conclusions in section 4 incl. Table 9. After reading through this work the message that the authors wish to deliver is not clear or justified at least to me. For example (as noted in the abstact) clearly name the effect of testing time in the final assessment of the specimens. Therefore I suggest to define in a more clear way what is to be communicated in the end. And of course connect it to the concept described in the Abstract or Intro as well as to literature findings of other researchers.
Author Response
We greatly appreciate your thoughtful comments that helped improve the manuscript. We have answered and responded to each question in paragraphs below.
1) This manuscript can be interesting to a readership involved in coatings evaluation. Several standards are cited, applied and results are reported. However in few cases a direct reference to the classification coding of the standards is not helpful to readers that are not familiar. For example reference to Coating grade according to ISO 2409 (Table 2) or class abbreviations in other Tables (i.e. Table 3) makes no sense if it is not defined or explained in a Supplementary Material.
Thank you for your feedback. For readers who are not familiar with the topics presented, we have included supplementary materials in which we define the environmental corrosivity classes according to ISO 12944-2/ISO 9223, present the degrees of adhesion of coatings according to ISO 2409 and the degrees of blistering, rusting, cracking and flaking according to ISO 4628-2÷5.
2) L19 please do not use abbreviations with no prior definition of meaning i.e. NSS and AASS.
We have expanded the abbreviations used to their full names.
3) L23 authors claim that the paper aims to introduce test for powder coating systems that are to be applied in corrosive conditions. How do the authors do that? Where in the text? Authors suggest to avoid the humid atmosphere containing SO2 and/or acidic salt spray tests. Please explain / justify.
To avoid confusion, the authors changed the word "introduce" to "presents" in the Abstract. This paper introduces (in the meaning of: presents/describes) a number of tests to which powder coatings are subjected (Table 1, L100 - L115, L129 - 159). Some of the accelerated laboratory tests focus on the aesthetic aspect of coatings (colour and gloss stability) and some focus on the corrosion resistance of the coatings and these types of tests are the subject of this article.
The results of the presented corrosion tests indicate that the applied test times for resistance to humid atmospheres containing SO2 (intermediate time of 240 h) and AASS (480 h) do not cause degradation of the tested powder coatings. Considering the probable specific working conditions of powder coatings, e.g. in industrial environments, it is not excluded that they will be exposed to acidic pH. Therefore, we acknowledge that it is an overinterpretation to state that these tests can be omitted. We have removed the corresponding sections from the text.
4) L23 typo error exists
Unfortunately, we did not find the error in the line provided and it has not been corrected.
5) L406 Authors suggest to “withdraw the test requirements like resistance to humid atmosphere containing SO2 and acidic salt spray“ for surfaces exposed to urban or road areas. How is this justified considering the SOx and NOx emissions by fossil fuel based combustion and burner units that exist? Thus the pH of precipitation is very often acidic. The authors need to provide more evidence that a salt spray test can substitute the chemical corrosion induced by the SOx or acid presence. I am not sure that this is feasible in the present manuscript thus I would suggest to re-write this part.
Thank you for pointing out an important issue. As we mention above, we applied excessive simplification in the text, and we acknowledge that our test results, which do not indicate degradation of the analysed coatings in humid atmospheres containing SO2 (intermediate time of 240 h) and acidic environments, are not the basis for such far-reaching conclusions. We have deleted sentences suggesting that it is not necessary to perform these tests.
6) L411-414 Authors claim that limited thickness of coatings is responsible for their low corrosion resistance. I believe this is a rather overismplistic approach as corrosion resistance is not just due to barrier properties.
Thank you for bringing this important issue to our attention. We certainly agree that the corrosion resistance of coatings, is related to both the thickness and adhesion as well as the protective mechanism. In the text, in a general way, we consider why there are different approaches to testing liquid and powder coatings as well as testing of powder coatings, but according to different requirements.
7) Several aging tests according to different quality standards have been performed in 2 types of coatings authors prepared. Results are nicely collected and presented and the overall technical features of the manuscript are good. However as a reader I do not get some solid conclusions in section 4 incl. Table 9. After reading through this work the message that the authors wish to deliver is not clear or justified at least to me. For example (as noted in the abstact) clearly name the effect of testing time in the final assessment of the specimens. Therefore I suggest to define in a more clear way what is to be communicated in the end. And of course connect it to the concept described in the Abstract or Intro as well as to literature findings of other researchers.
The paper does not focus on the failure mechanisms of the powder coatings studied, but only on highlighting the differences between the quality requirements. Hence, it is perhaps a mistaken impression that the paper ignores the literature findings of other researchers. This will be the subject of a future article where we will present the results of the intermediate evaluations and EIS analysis in more detail, and we will certainly benefit from the valuable suggestions in this review. In this publication in a simple way we wanted to present the differences in the required corrosion test times that are not fully understood and, as the presented results show, not fully justified.
The key part of the abstract is:
„ Among European quality certificates for powder coatings applied to galvanized steel, the most commonly recognized are GSB-ST and Qualisteelcoat certificates, which also refer to the EN 13438 standard. […] This paper presents an experimental evaluation of how the required length of selected accelerated corrosion tests can affect the final assessment of powder coatings.”
It seems to us that Section 4 and especially the following part adequately represents the content of the article:
„The paper presents the results of accelerated aging tests of selected powder coatings performed according to different quality standards: GSB-ST 663-4, Qualisteelcoat and EN 13438. A summary of the tests results and their evaluation to the criteria of the discussed quality requirements is presented in Table 9. The differences in the evaluation criteria and required accelerated corrosion test durations are so significant that the tested two-coat epoxy-polyester (EP+P) system applied to continuously galvanized steel, despite its good corrosion resistance confirmed with impedance level, would only have a chance of achieving a positive evaluation according to the quality requirements of EN 13438 (assuming it passed the other mechanical and weathering tests). […]. However, it must be considered that behind each powder coating quality certificate, different quality requirements are specified by a given standard. This paper presents a practical example of how differences in the required length of selected accelerated corrosion tests can affect the final assessment of powder coatings.”
Accelerated chamber tests are a partial, but important, section of the testing that powder coatings are subjected to. Unfortunately, we cannot further simplify Table 9, which summarises all accelerated corrosion test results obtained. It seems to us that the Table 9 clearly indicates the differences in test times and the fact whether they are required at all according to the different European quality recommendations and EN 13438 standard, which was the subject of the article.
Round 2
Reviewer 1 Report
Please revise it by following the previous comments. It is not an excuse to decline the revision by allocating these necessary data to a new journal paper.
(1) Please retake pictures for Fig.2 and 3 to avoid light reflection from the sample.
(2) For Fig.3, it is better to provide the macroscopy of polyester coating after tests for different times for comparison.
(3) In table 2, it seems that both coatings become thicker after the test, about 10 um. Please explain this result.
(4) Please provide EIS along with equivalent circuits besides the Bode.
Author Response
Thank you for the time you dedicated to reviewing our publication and all your comments. You will find the answers to your questions in the attached file.
Reviewer 2 Report
no comment
Author Response
Thank you again for your valuable professional feedback that helped us improve our publication.